

# A comparison of species composition and community assemblage of secondary forests between the birch and pine-oak belts in the mid-altitude zone of the Qinling Mountains, China

Zongzheng Chai and  Dexiang Wang

College of Forestry, Northwest A&F University, Yangling, Shaanxi, People's Republic of China

## ABSTRACT

The mid-altitude zone of the Qinling Mountains in China was once dominated by birch and pine-oak belts but are now mainly covered by secondary growth following large-scale deforestation. Assessing the recovery and sustainability of these forests is essential for their management and restoration. We investigated and compared the tree species composition and community assemblages of secondary forests of the birch and pine-oak belts in the Huoditang forest region of the Qinling Mountains after identical natural recoveries. Both types of belts had rich species compositions and similar floristic components but clearly different community structures. Tree diversity was significantly higher for the birch than the pine-oak belt. Niche and neutral processes simultaneously influenced the species distribution and community dynamics of the belts, and these forests were able to maintain stable development during natural recoveries. The conservation and management of these forests should receive more attention to protect biodiversity and the forest resources in the Qinling Mountains.

## INTRODUCTION

Conserving biodiversity in forests has long been an important global concern (*Brockerhoff et al., 2008*; *Ratcliffe et al., 2015*), because forest ecosystems provide services essential to human well-being and refuges for terrestrial plants and animals (*Schuldt & Scherer-Lorenzen, 2014*; *Sharma et al., 2010*). Rapid changes in forest landscapes due to urbanization, agriculture, road construction, and especially deforestation have caused forest loss and fragmentation, threatening forest biodiversity worldwide (*Elliott & Swank, 1994*; *Imai et al., 2014*; *Jung et al., 2014*). Urgent intervention for conserving biodiversity and forest remnants is thus necessary (*Jactel & Brockerhoff, 2007*; *Nyafwono et al., 2014*; *Oishi & Doei, 2015*).

Large areas of primary forest in China were cut between the 1950s and 1980s. After a long period of recovery, secondary forests formed with varying patterns of natural succession (*Kan, Wang & Wu, 2015*), which now account for approximately 50% of the

Corresponding author
Dexiang Wang, wangdx66@126.com, wangdx66@sohu.com

total forested area in China (*Chen, Zhou & Zhu, 1994*; *Yan, Zhu & Gang, 2013*; *Yang, Shi & Zhu, 2013*). Forest restoration has been increasingly addressed by the Chinese government and ecologists, because deforestation has caused serious environmental problems and the loss of ecological services (*Huang et al., 2006*; *Zhang et al., 2010*).

The Qinling Mountains are speciose and a key region of biodiversity of global importance. The forests in the mountains unfortunately suffered from large-scale deforestation in the 1960s and 1970s. Young secondary forests now cover large areas and increasingly define the prospects of long-term conservation of ecosystemic services and biodiversity (*Cheng et al., 2015*; *Wang et al., 2015*). The mid-altitude zone covers a large area, with complicated geomorphology and various climatic and soil conditions, and is characterized by the richest species diversity in the Qinling Mountains. Birch (*Betula*) and pine-oak (*Pinus-Quercus*) belts are the two main types of vegetation in the zone (Fig. 1) (*Liu et al., 2001*; *Zhao, Ma & Xiao, 2014*) and play important roles in the establishment and maintenance of ecosystems and their functions, such as the conservation of soil and water (*Chai & Wang, 2015*; *Lei et al., 1996a*; *Lei et al., 1996b*).

Previous studies have determined that the deforestation led to variation in the landscape pattern of the secondary forests (*Lei et al., 1996a*; *Lei et al., 1996b*; *Wang et al., 2015*; *Zhang et al., 2014*), community dynamics (*Chai & Wang, 2015*; *Ma et al., 2014*; *Zhang et al., 2014*), regeneration characteristics (*Chai & Wang, 2015*; *Yu et al., 2013*), nutrient cycles (*Liu et al., 2001*), and soil properties (*Cheng et al., 2013*; *Cheng et al., 2015*; *Ren et al., 2012*) in the Qinling Mountains. These studies, however, did not compare the characteristics of the birch and pine-oak belts, especially the secondary vegetation that established at the same time and region after clear-cutting. We investigated and compared the tree species composition and community assemblages of secondary forests in the birch and pine-oak belts in the Huoditang forest region of the Qinling Mountains after identical natural recoveries. The following questions were addressed: (1) How do species composition and community structure vary among different secondary forest types? (2) Do the patterns of tree diversity in the secondary forests differ between the birch and pine-oak belts? (3) What are the underlying ecological mechanisms for the community assemblages of the secondary forests in the mid-altitude zone of the Qinling Mountains, China? We aimed to improve our understanding of the status of secondary forests and to contribute to the success of vegetation restoration and the conservation of biodiversity.

## MATERIALS AND METHODS

### Study site

The Qinling Mountains are in the transitional region between the subtropical and warm temperate zones of central China and are generally considered as the physical geographical dividing line between southern and northern China. The mountains are valuable reservoirs of biodiversity and play a key role in the maintenance of other natural resources, such as soils, air, and water. The vegetation of, and environmental change in, the mountains have long been of academic interest due to the unique geographical location (*Dang et al., 2010*; *Huang et al., 2006*; *Wang et al., 2015*; *Zhang et al., 2013*). The vegetation displays a vertical
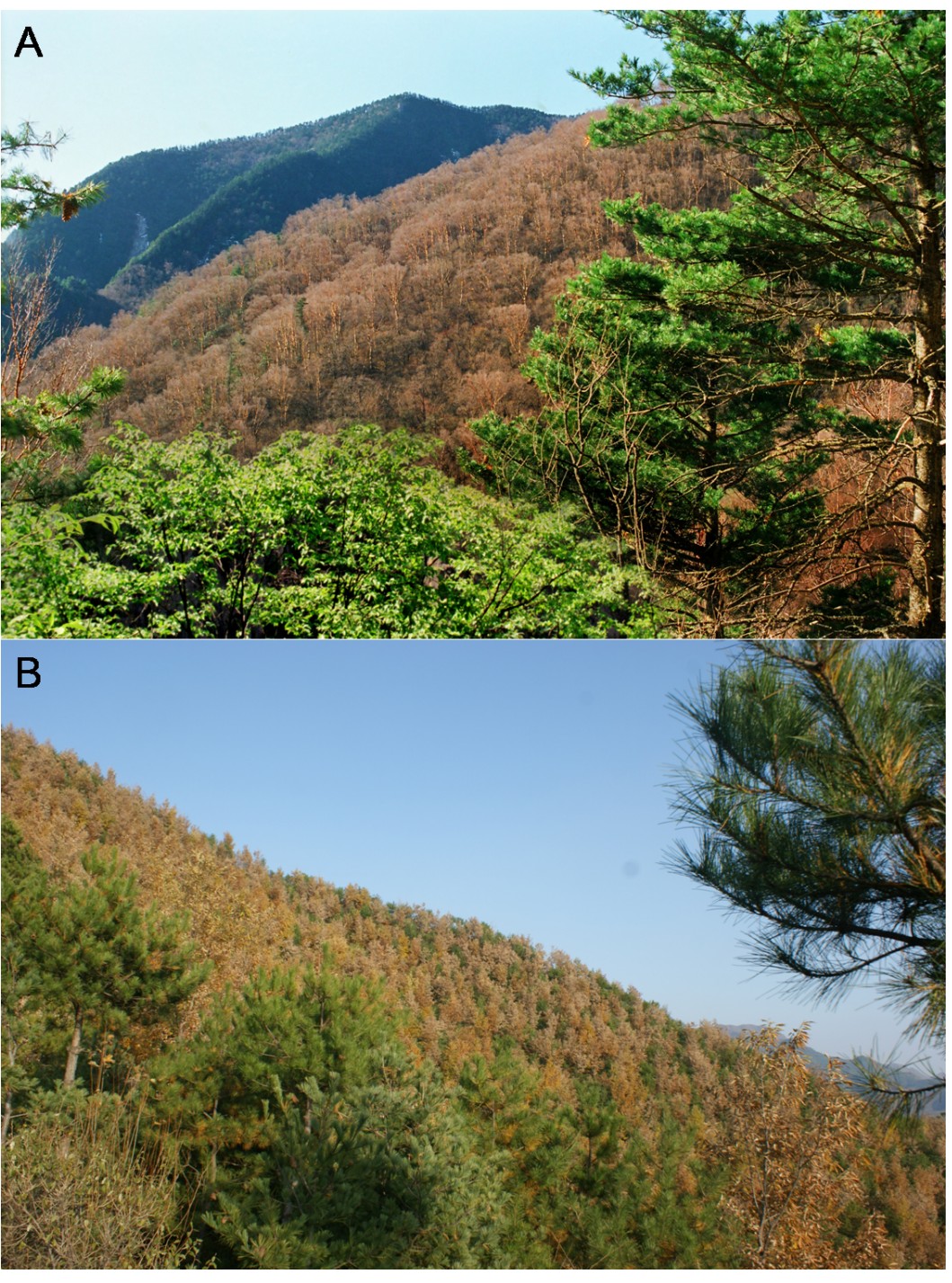

**Figure 1** **Birch (A) and pine-oak (B) belts in the mid-altitude zone of the Qinling Mountains, China.** Photos taken on October 2012.

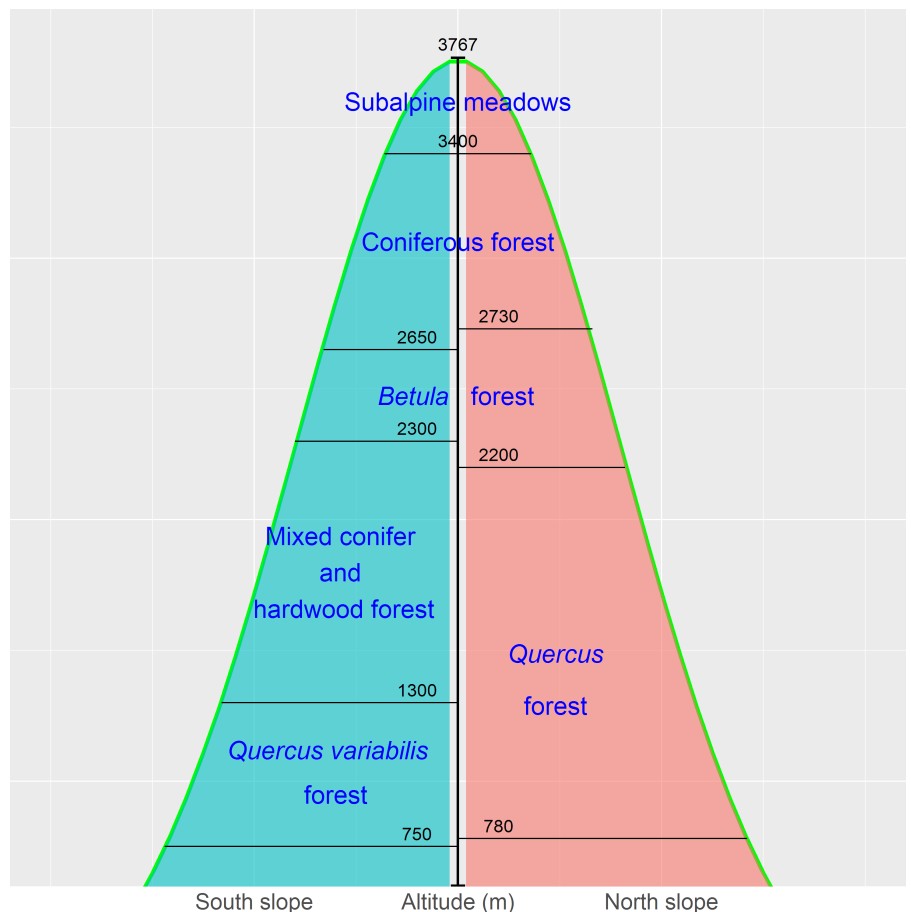

**Figure 2** The vertical zones of vegetation in the Qinling Mountains, China (*Zhao, Ma & Xiao, 2014*).

zonation. The zones in Fig. 2 represent a general model for Taibai Mountain, the highest peak in the Qinling Mountains, with a summit altitude of 3767 m a.s.l. The zones extend laterally and vary locally (*Fang & Gao, 1963*; *Zhao, Ma & Xiao, 2014*).

The birch belt at 2,200–2,700 m contains *Betula albosinensis* Burk., *B. utilis*D. Don, *B. luminifera* H. Winkl., and *B. platyphylla* Suk. Pine-oak mixed forests and mosaic pure forests of *Pinus tabuliformis* Carr., *P. armandii* Franch., and *Quercus aliena* var. *acutiserrata* Maxim. are distributed at 800–2,300 m and constitute the pine-oak belt (*Liu et al., 2001*; *Zhao, Ma & Xiao, 2014*; *Wang et al., 2015*). These two forest belts are the most common types in the mid-altitude zone (1,300–2,600 m) of the Qinling Mountains.

We conducted a field survey at the Qinling National Forest Ecosystem Research Station in the Huoditang forest region in Ningshan County. The Huoditang forest region at 850–2,470 m is in the typical vertical vegetation zone on the south slopes of the Qinling Mountains, and the research station is in the mid-altitude zone between 1,400 and 2,400 m. The birch belt is distributed at higher elevations of the mid-altitude zone (1,800–2,400 m), and the pine-oak belt is widely distributed at lower elevations (1,300–2,000 m) (*Wang et al., 2015*).

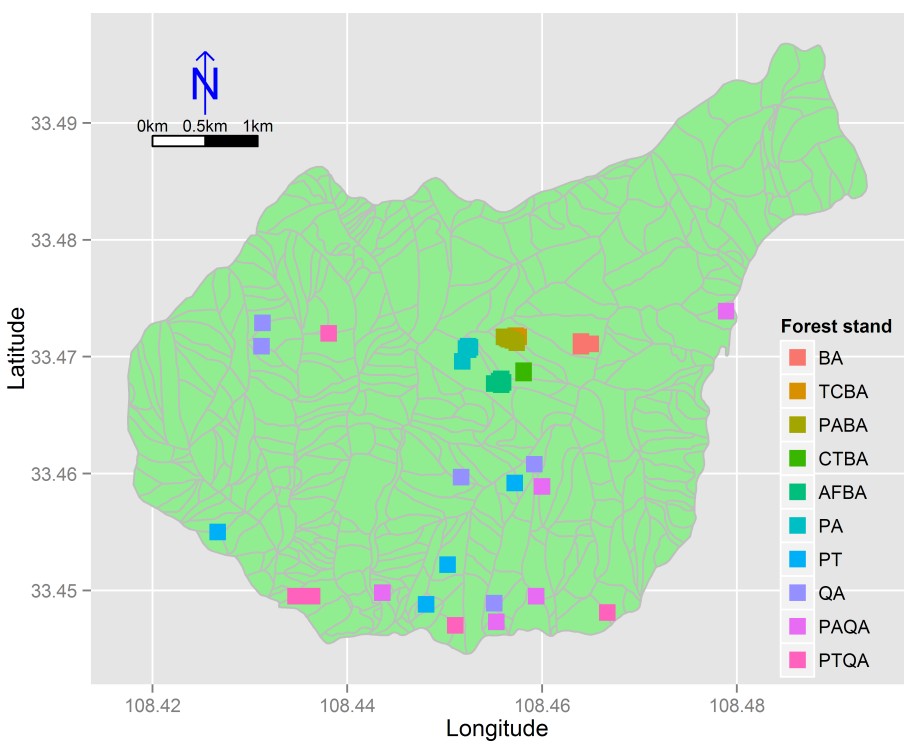

**Figure 3  The distribution of the 50 sample plots in 10 forest stands of the birch and pine-oak belts in the mid-altitude zone of the Huoditang forest region.** The polygons are the distribution of sub-compartment. See Table 1 for the stand codes.

Most areas of the Huoditang forest region were last deforested during the 1960s and 1970s, and 95% of the area are now covered by secondary growth (*Cheng et al., 2013*; *Lei, Peng & Chen, 1996*). The forest region has rich plant resources and complex forest types, and the area of secondary forest is large and centrally distributed. The Huoditang forest was thus favorable for studying the secondary forests in the Qinling Mountains (*Chai & Wang, 2015*; *Cheng et al., 2013*; *Lei, Peng & Chen, 1996*; *Wang et al., 2015*).

## Field sampling

We divided the birch and pine-oak belts into five forest types (Table 1) based on a previous study (*Lei, Peng & Chen, 1996*) and a reconnaissance survey. These forest types are the most common in the mid-altitude zone of the Huoditang forest region. A total of 50 permanent plots (30 × 30 m) were established, 25 plots for each of the birch and pine-oak belts, and data were collected from July to September in 2012–2014 using typical sampling methods for surveying the floristic composition, diversity, and structure of the forests (Fig. 3). Five plots were randomly placed in each of the five forest types in each of the birch and pine-oak belts. The total study area was 4.5 ha. The elevation, slope, aspect, and GPS location of each plot were determined. The forest types met the following criteria: (1) stand age of approximately 50–60 years, representing the earliest and largest secondary forests after the deforestations; (2) minimal disturbance after cutting; and (3) similar habitat conditions among the forest types.

**Table 1** Main forest types of the birch and pine-oak belts in the mid-altitude zone of the Huoditang forest region of the Qinling Mountains, China.

| Forest belt | Forest stand | Code |
|---|---|---|
| Birch | *Betula albosinensis* | BA |
| | *Tsuga chinensis* + *Betula albosinensis* | TCBA |
| | *Pinus armandii* + *Betula albosinensis* | PABA |
| | *Carpinus turczaninowii* + *Betula albosinensis* | CTBA |
| | *Abies fargesii* + *Betula albosinensis* | AFBA |
| Pine-oak | *Pinus armandii* | PA |
| | *Pinus tabuliformis* | PT |
| | *Quercus aliena* var. *acutiserrata* | QA |
| | *Pinus armandii* + *Quercus aliena* var. *acutiserrata* | PAQA |
| | *Pinus tabuliformis* + *Quercus aliena* var. *acutiserrata* | PTQA |

All trees with a diameter at breast height (DBH, at 1.3 m) $\geq$ 5 cm were marked, and their locations were determined using a total station (TOPCON-GTS-602AF). Canopy closure, stem height (height of the first major branch), tree height, DBH, crown width, and health status were surveyed for the trees in each plot. This work was conducted based on Forestry Standards "Observation Methodology for Long-term Forest Ecosystem Research" of the People's Republic of China (LY/T 1952–2011).

## Data analysis
### Importance values (IVs)
The importance value ($IV$) of a species is defined as the average of its relative density ($RD$), relative frequency ($RF$), and relative dominance ($Rd$). The IVs of the tree species were calculated as (*Arbainsyah, Kustiawan & De Snoo, 2014*; *Curtis & McIntosh, 1951*):

$$\text{Density}(D) = \frac{\text{Number of individuals of a species}}{\text{Area of all sample units}}$$

$$\text{Relative abundance}(RD) = \frac{\text{Number of individuals of a species}}{\text{Density for all species}} \times 100\%$$

$$\text{Frequency}(F) = \frac{\text{Number of quadrats containing a certain specis}}{\text{Total number of quadrats}}$$

$$\text{Relative Frequency}(RF) = \frac{\text{Frequency of a certain species}}{\text{Total number of species}} \times 100\%$$

$$\text{Dominance}(d) = \frac{\text{Basal area of a species}}{\text{Area of all sample units}}$$

$$\text{Relative Dominance}(Rd) = \frac{\text{Dominance of one specis}}{\text{Domiance of all species}} \times 100\%$$

$$IV = (RD + RF + Rd)/3.$$

### Rarefaction and extrapolation curves with Hill numbers
Hill numbers, or the effective number of species, are a mathematically unified family of diversity indices (differing among themselves only by an exponent $q$) (*Hill, 1973*), and are increasing used to characterize the taxonomic, phylogenetic, or functional diversity of an

assemblage (*Chao, Chiu & Jost, 2010*; *Chao, Chiu & Jost, 2014*; *Chao et al., 2014*; *Chiu, Jost & Chao, 2014*; *Ibanez, Grytnes & Birnbaum, 2016*). Integrated curves based on sampling theory that smoothly link rarefaction and extrapolation standardize samples on the basis of sample size or sample completeness and facilitate the comparison of biodiversity data. To characterize the species diversity of an assemblage, *Chao et al. (2014)* applied a unified approach for both individual-based data and sample-based data to estimate rarefaction and extrapolation curves for the first three Hill numbers: species richness ($q = 0$), the exponential of the Shannon entropy (Shannon diversity, $q = 1$), and the inverse Simpson concentration (Simpson diversity, $q = 2$). The proposed estimators are accurate for both rarefaction and short-range extrapolation.

We compared the patterns of species diversity using the rarefaction and extrapolation curves with Hill numbers. Constructing rarefied and extrapolated curves produced patterns based on abundance and incidence data, respectively. Species diversity (species richness, Shannon diversity, and Simpson diversity) was estimated as the mean of 200 bootstrap replications with 95% confidence intervals.

### Cluster and correspondence analyses

The similarities in the species compositions and distributional patterns were explored using clustering and correspondence analyses. The objective of a clustering analysis is to identify subgroups within a group. Clustering analysis generally refers to the methods that attempt to categorize the data into subgroups such that the observations within the same group are more similar compared to the observations in different groups (*Legendre & Legendre, 2012*). Correspondence analysis is a powerful method for the multivariate exploration of large-scale data (*Greenacre, 1984*), which are commonly used by ecologists to analyze data on the incidence or abundance of species in samples (*Cakir, Khorram & Nelson, 2006*; *Hill, 1974*), and provides a robust statistical tool for understanding species distribution relative to environmental factors (*Beebe et al., 2000*; *Ter Braak, 1985*). We used cluster analysis with group averages based on the species-abundance data of the forest stands, categorizing the ten forest stands into two major groups, and then applied the correspondence analysis to exploit the same information as used in the cluster analysis to strengthen the validity of the cluster analysis.

### Species abundance distribution (SAD)

The following six SAD models were considered: broken-stick, niche-preemption, log-normal, Zipf, Zipf-Mandelbrot, and neutral-theory models (Table 2, for the details of these models, see the introduction in Supplemental Information 3). The Kolmogorov–Smirnov (K-S) test was applied for comparing the discrepancy of the fitted and observed SAD patterns; this test is recommended for testing the agreement to models of abundance distribution (*Hill & Hamer, 1998*; *Basset et al., 1998*) because it is more powerful than the chi-square test. The Akaike Information Criterion (AIC) method was also used to compare the models and identify the best model by using log-likelihoods (log L) of the fitted models as the input (*Filho, Martins & Gneri, 2002*). AIC is calculated by:

$$AIC = -2 \log L + 2k$$

where $k$ is the parameter number in the fitted model.

**Table 2** Six main models for the distribution of species abundance.

| Model | Equation | Code | Reference |
|---|---|---|---|
| Broken-stick | $\hat{a}_r = \frac{N}{S}\sum_{k=r}^{S}\frac{1}{k}$ | (1) | *MacArthur (1957)* |
| Niche-preemption | $\hat{a}_r = N\alpha(1-a)^{r-1}$ | (2) | *Motomura (1932)* |
| Log-normal | $\hat{a}_r = \exp[\log(u)+\log(\sigma)\Phi]$ | (3) | *Preston (1948)* |
| Zipf | $\hat{a}_r = N\hat{p}_1 r^{\gamma}$ | (4) | *Frontier (1987)* |
| Zipf-Mandelbrot | $\hat{a}_r = Nc(r+\beta)^{\gamma}$ | (5) | |
| Neutral-theory | $\phi_n = \theta\frac{J!}{n!(J-n)!}\frac{\Gamma(\gamma)}{\Gamma(J+\gamma)}\int_0^{\gamma}\frac{\Gamma(n+y)}{\Gamma(1+y)}\frac{\Gamma(J-n+\gamma-y)}{\Gamma(\gamma-y)}\exp(-y\theta/\gamma)dy$ | (6) | *Hubbell (2001)* |

**Notes.**

$\hat{a}_r$, expected abundance of species of rank $r$; $S$, number of species; $N$, number of individuals; $\Phi$, a standard normal function; $\hat{p}_1$, estimated proportion of the most abundant species; $\alpha, \sigma, \gamma, \beta$, and $c$, estimated parameters in each model. For the neutral-theory model, $\Gamma(z) = \int_0^{\infty}t^{z-1}e^{-t}dt$, which is equal to $(z-1)!$, for integer $z$, $\gamma = \frac{m(J-1)}{1-m}$, $\theta$ is a fundamental diversity number, and $m$ is the migration rate.

### Statistical analyses

R version 3.1.3 (*R Core Team, 2015*) was used for all statistical analyses. Cluster analysis, correspondence analysis, and SAD were conducted using the vegan (*Oksanen et al., 2008*) and untb (*Robin, 2009*) packages. Rarefaction and extrapolation curves were compiled using the iNEXT package (*Chao et al., 2014*). The figures were drawn and the data were manipulated using the ggplot2 (*Hadley, 2015*) and reshape2 (*Hadley, 2014*) packages, respectively.

## RESULTS

### Tree species composition

A total of 50 tree species belonging to 30 genera in 16 families were identified among 5,686 individual trees (DBH $\geq$ 5 cm) in the 50 plots (totaling 4.5 hm$^2$) from the 10 typical secondary forest stands in the two forest belts in the mid-altitude zone of the Qinling Mountains. The attributes of the stands are summarized in Table 3. The 25 plots of the birch belt contained 2,934 individual trees in 43 species (27 genera, 16 families). The 25 plots of the pine-oak belt contained 2752 individual trees in 41 species (28 genera, 14 families) (Table 3 and the species composition and IV characteristics in Supplemental Information 4).

Four species, *Q. aliena* var. *acutiserrata*, *P. armandii*, *Toxicodendron vernicifluum* (Stokes) F. A. Barkl., and *Carpinus turczaninowii* Hance had the broadest distributions, irrespective of forest type. The dominant species in the birch belt were *B. albo sinensis* (IV = 10.63%), *P. armandii* (10.19%), *Acer davidii* Franch. (8.76%), and *T. vernicifluum* (8.25%). The dominant species in the pine-oak belt were *Q. aliena* var. *acutiserrata* (26.15%), *P. tabuliformis* (22.50%), *P. armandii* (20.05%), and *T. vernicifluum* (10.27%) (See the species composition and IV characteristics in Supplemental Information 4).

The seven most common families were Pinaceae, Fagaceae, Aceraceae, Betulaceae, Anacardiaceae, Rosaceae, and Lauraceae. These families accounted for 91.44% of all trees recorded and were among the ten most important families in both the birch and pine-oak belts. Aceraceae, Pinaceae, and Betulaceae were the dominant families with the highest

Chai and Wang (2016), *PeerJ*, DOI 10.7717/peerj.1900

Peerj

**Table 3** Summary of the stand attributes of the typical secondary forests in the mid-altitude zone of the Qinling Mountains, China. See Table 1 for the stand codes.

| Item | | Forest stand | | | | | | | | | | Forest belt | |
|------|------|------|------|------|------|------|------|------|------|------|------|------|------|
| | | BA | TCBA | PABA | CTBA | AFBA | PA | PT | QA | PAQA | PTQA | Birch | Pine-oak |
| Sample number | | 5 | 5 | 5 | 5 | 5 | 5 | 5 | 5 | 5 | 5 | 25 | 25 |
| Forest area (m$^2$) | | 4,500 | 4,500 | 4,500 | 4,500 | 4,500 | 4,500 | 4,500 | 4,500 | 4,500 | 4,500 | 22,500 | 22,500 |
| Stand age (a) | | 50–60 | 50–60 | 50–60 | 50–60 | 50–60 | 50–60 | 50–60 | 50–60 | 50–60 | 50–60 | 50–60 | 50–60 |
| Family number | | 13 | 12 | 13 | 11 | 10 | 13 | 10 | 10 | 10 | 8 | 16 | 14 |
| Genera number | | 22 | 19 | 20 | 17 | 16 | 18 | 16 | 13 | 17 | 14 | 27 | 28 |
| Species number | | 32 | 27 | 32 | 25 | 25 | 24 | 22 | 17 | 22 | 17 | 43 | 41 |
| Diameter at breast height (cm) | min | 13.04 | 13.32 | 14.06 | 16.70 | 13.90 | 17.87 | 14.06 | 16.27 | 14.94 | 14.77 | 13.04 | 14.06 |
| | max | 16.94 | 17.18 | 16.12 | 19.06 | 17.24 | 21.44 | 19.42 | 22.23 | 20.08 | 21.04 | 19.06 | 22.23 |
| | mean | 14.77 | 14.43 | 15.24 | 18.00 | 15.26 | 19.27 | 16.37 | 18.68 | 16.81 | 17.36 | 15.54 | 17.70 |
| Tree height (m) | min | 8.46 | 10.89 | 9.75 | 10.08 | 12.14 | 17.25 | 10.39 | 10.39 | 10.18 | 12.87 | 8.46 | 10.18 |
| | max | 10.44 | 11.81 | 14.82 | 11.01 | 16.56 | 20.21 | 19.17 | 19.04 | 16.11 | 19.48 | 16.56 | 20.21 |
| | mean | 9.54 | 11.18 | 12.09 | 10.51 | 14.91 | 19.13 | 13.66 | 13.59 | 13.57 | 16.22 | 11.65 | 15.23 |
| Crown width (m) | min | 0.60 | 0.70 | 0.70 | 0.55 | 0.00 | 0.70 | 0.50 | 0.50 | 1.30 | 1.65 | 0.00 | 0.50 |
| | max | 11.35 | 11.40 | 15.05 | 8.65 | 10.15 | 11.75 | 9.50 | 9.60 | 10.05 | 16.85 | 15.05 | 16.85 |
| | mean | 3.98 | 3.96 | 4.44 | 3.91 | 4.44 | 4.20 | 3.30 | 4.18 | 4.90 | 5.51 | 4.17 | 4.36 |
| Basal area (m$^2$ ha$^{-1}$) | min | 29.44 | 24.50 | 30.20 | 21.54 | 26.34 | 24.48 | 32.82 | 31.82 | 21.13 | 22.91 | 21.54 | 21.13 |
| | max | 37.31 | 33.25 | 45.44 | 30.70 | 32.40 | 37.05 | 46.78 | 64.16 | 38.09 | 43.87 | 45.44 | 64.15 |
| | mean | 31.99 | 27.28 | 34.80 | 26.27 | 30.02 | 30.98 | 40.36 | 43.41 | 30.44 | 36.10 | 30.07 | 36.26 |
| Stand density (trees ha$^{-1}$) | min | 1,122 | 967 | 1,400 | 800 | 944 | 767 | 1,156 | 1,167 | 822 | 1,067 | 800 | 767 |
| | max | 1,867 | 1,478 | 2,100 | 878 | 1,411 | 1,189 | 1,789 | 1,789 | 1,456 | 1,356 | 2,100 | 1,789 |
| | mean | 1,511 | 1,345 | 1,593 | 835 | 1,235 | 929 | 1,493 | 1,385 | 1,073 | 1,236 | 1,304 | 1,223 |

**Table 4** Ten most important tree families, in descending order of overall relative importance (ORI), for the birch and pine-oak belts in the mid-altitude zone of the Qinling Mountains, China.

| Rank | Birch belt | R.Ab | R.Fr | ORI | Pine-oak belt | R.Ab | R.Fr | ORI |
|---|---|---|---|---|---|---|---|---|
| 1 | Aceraceae | 23.59 | 11.31 | 34.9 | Pinaceae | 46.84 | 17.24 | 64.08 |
| 2 | Pinaceae | 19.39 | 11.31 | 30.7 | Fagaceae | 33.68 | 15.17 | 48.85 |
| 3 | Betulaceae | 15.78 | 11.31 | 27.09 | Anacardiaceae | 6.8 | 13.79 | 20.59 |
| 4 | Rosaceae | 12.07 | 11.31 | 23.38 | Betulaceae | 3.85 | 11.72 | 15.57 |
| 5 | Anacardiaceae | 7.74 | 10.41 | 18.15 | Lauraceae | 2.18 | 8.97 | 11.15 |
| 6 | Fagaceae | 7.53 | 10.41 | 17.94 | Cornaceae | 1.89 | 7.59 | 9.48 |
| 7 | Salicaceae | 6.95 | 5.88 | 12.83 | Juglandaceae | 1.53 | 5.52 | 7.05 |
| 8 | Lauraceae | 1.87 | 7.24 | 9.11 | Aceraceae | 1.13 | 5.52 | 6.65 |
| 9 | Araliaceae | 2.22 | 5.88 | 8.1 | Tiliaceae | 0.69 | 3.45 | 4.14 |
| 10 | Bignoniaceae | 1.64 | 4.98 | 6.62 | Rosaceae | 0.65 | 3.45 | 4.1 |
| | $\sum 1-10$ | 98.78 | 90.04 | 188.82 | $\sum 1-10$ | 99.24 | 92.42 | 191.66 |
| | $\sum 11-16$ | 1.22 | 9.95 | 11.17 | $\sum 11-14$ | 0.76 | 7.59 | 8.35 |

**Notes.**
R.Ab, relative abundance; R.Fr, relative frequency.

**Table 5** Ten most important tree genera, in descending order importance (ORI), for the birch and pine-oak belts in the mid-altitude zone of the Qinling Mountains, China.

| Rank | Birch belt | R.Ab | R.Fr | ORI | Pine-oak belt | R.Ab | R.Fr | ORI |
|---|---|---|---|---|---|---|---|---|
| 1 | Acer | 23.59 | 7.55 | 31.14 | Pinus | 44.33 | 14.12 | 58.45 |
| 2 | Betula | 10.02 | 7.55 | 17.57 | Quercus | 33.68 | 12.43 | 46.11 |
| 3 | Pinus | 8.52 | 7.55 | 16.07 | Toxicodendron | 6.58 | 11.3 | 17.88 |
| 4 | Sorbus | 8.45 | 7.55 | 16 | Carpinus | 2.58 | 7.34 | 9.92 |
| 5 | Toxicodendron | 7.74 | 6.95 | 14.69 | Lindera | 1.85 | 6.21 | 8.06 |
| 6 | Tsuga | 7.6 | 6.95 | 14.55 | Juglans | 1.27 | 4.52 | 5.79 |
| 7 | Quercus | 7.53 | 6.95 | 14.48 | Acer | 1.13 | 4.52 | 5.65 |
| 8 | Carpinus | 3.99 | 6.34 | 10.33 | Tsuga | 1.09 | 3.95 | 5.04 |
| 9 | Cerasus | 3.61 | 5.74 | 9.35 | Betula | 0.76 | 3.95 | 4.71 |
| 10 | Populus | 5.42 | 2.72 | 8.14 | Larix | 1.16 | 2.82 | 3.98 |
| | $\sum 1-10$ | 86.47 | 65.85 | 152.32 | $\sum 1-10$ | 94.43 | 71.16 | 165.59 |
| | $\sum 11-27$ | 13.53 | 34.12 | 47.65 | $\sum 11-28$ | 5.56 | 28.77 | 34.33 |

**Notes.**
R.Ab, relative abundance; R.Fr, relative frequency.

values of overall relative importance (ORI) in the birch belt. Pinaceae, Fagaceae, and Anacardiaceae were the dominant families in the pine-oak belt (Table 4).

*Acer, Betula, Pinus, Toxicodendron, Tsuga, Quercus,* and *Carpinus* were among the most common and important genera in both forest belts. *Acer, Betula,* and *Pinus* were the dominant genera with the highest ORIs in the birch belt. *Pinus, Quercus,* and *Toxicodendron* were the dominant genera in the pine-oak belt (Table 5).

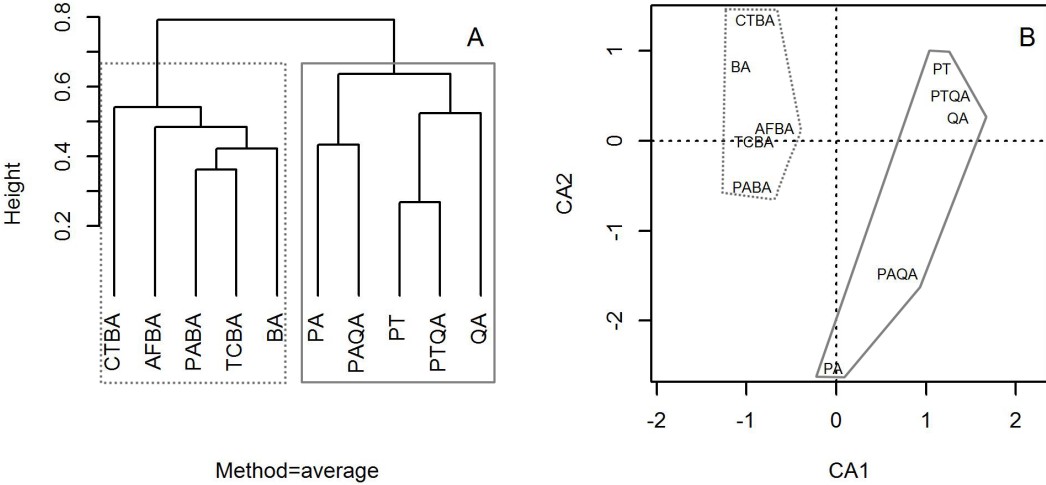

**Figure 4 Dendrogram from the cluster analysis based on group averages (A) and correspondence analysis ordination diagram (B) of the 10 typical secondary forests in the birch and pine-oak belts in the mid-altitude zone of the Qinling Mountains, China.** The grey dotted polygon is the birch belt and the grey solid polygon is the pine-oak belt. See Table 1 for the stand codes.

## Similarity among tree community structures

Cluster analysis with group averages based on the species composition and abundance of forest stands divided the ten forest stands into two major groups, corresponding to the birch and pine-oak belts (Fig. 4). Correspondence analysis ordination supported the findings of the cluster analysis, indicating similarities and differences among the ten forest stands.

## Comparison of tree species diversity in the two forest belts based on abundance data

We constructed individual-based and coverage-based rarefaction and extrapolation curves for Hill numbers $q = 0$, 1, and 2 to compare the diversities of the birch and pine-oak forest belts (Fig. 5). The reference sample size (number of individual trees) for the birch belt was 2,934, and observed species richness ($q = 0$), Shannon diversity ($q = 1$), and Simpson diversity ($q = 2$) for this reference sample size were 43, 23, and 17.72, respectively. The reference sample size for the pine-oak belt was 2,752, and the observed species richness, Shannon diversity, and Simpson diversity were 41, 7.51, and 4.63, respectively.

We extrapolated the reference sample size to 5,504 (double the smaller reference sample size), and the base coverage (the lowest coverage for the doubled reference sample sizes or the maximum coverage for reference samples, whichever was larger) was closer to 1.0. Both individual-based and coverage-based rarefaction and extrapolation curves indicated that the birch belt was more diverse than the pine-oak belt, although the confidence intervals for species richness overlapped.

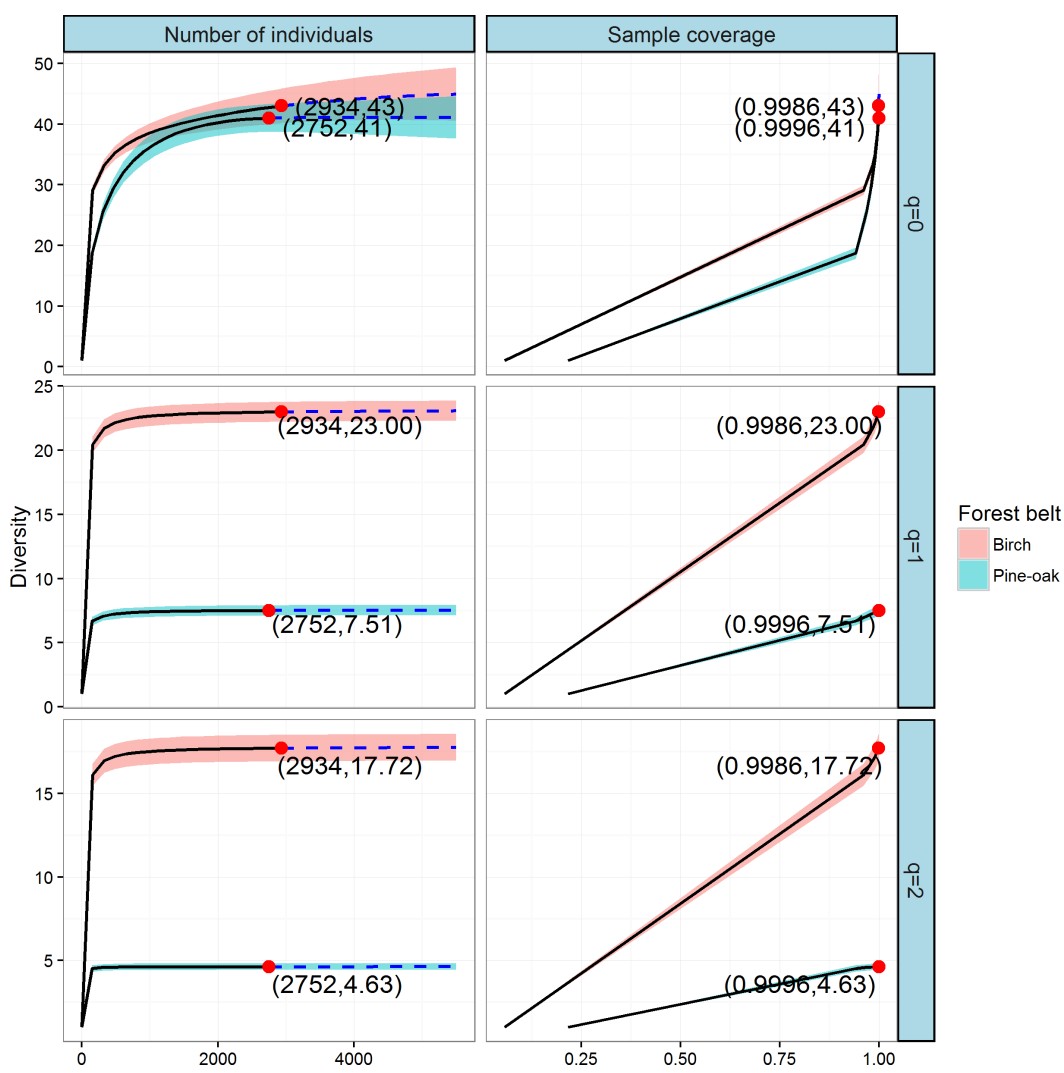

**Figure 5 Individual-based and coverage-based rarefaction and extrapolation curves based on the Hill numbers ($q = 0, 1, 2$) for the birch and pine-oak belts.** The 95% confidence intervals (shaded regions) were obtained by a bootstrap method based on 200 replications. Reference samples are denoted by solid dots, the numbers in the parentheses are the sample size and the observed Hill number for each reference sample.

## Comparison of tree species diversity in the two forest belts based on incidence data

We next constructed the sample-based and coverage-based rarefaction and extrapolation curves for Hill numbers $q = 0, 1, 2$ to compare the diversities of the birch and pine-oak forest belts (Fig. 6). Both belts had the same sample size (25) and species richness (25). The Shannon diversities were 24.82 and 23.53 and the Simpson diversities were 24.65 and 22.24 for the birch and pine-oak forest belts, respectively. We extrapolated the reference sample size to 50 (double the smaller reference sample size) and the base coverage to 1.0, indicating that sampling was nearly complete for these two belts. Both the sample-based and coverage-based rarefaction curves indicated little overlap between the Shannon and

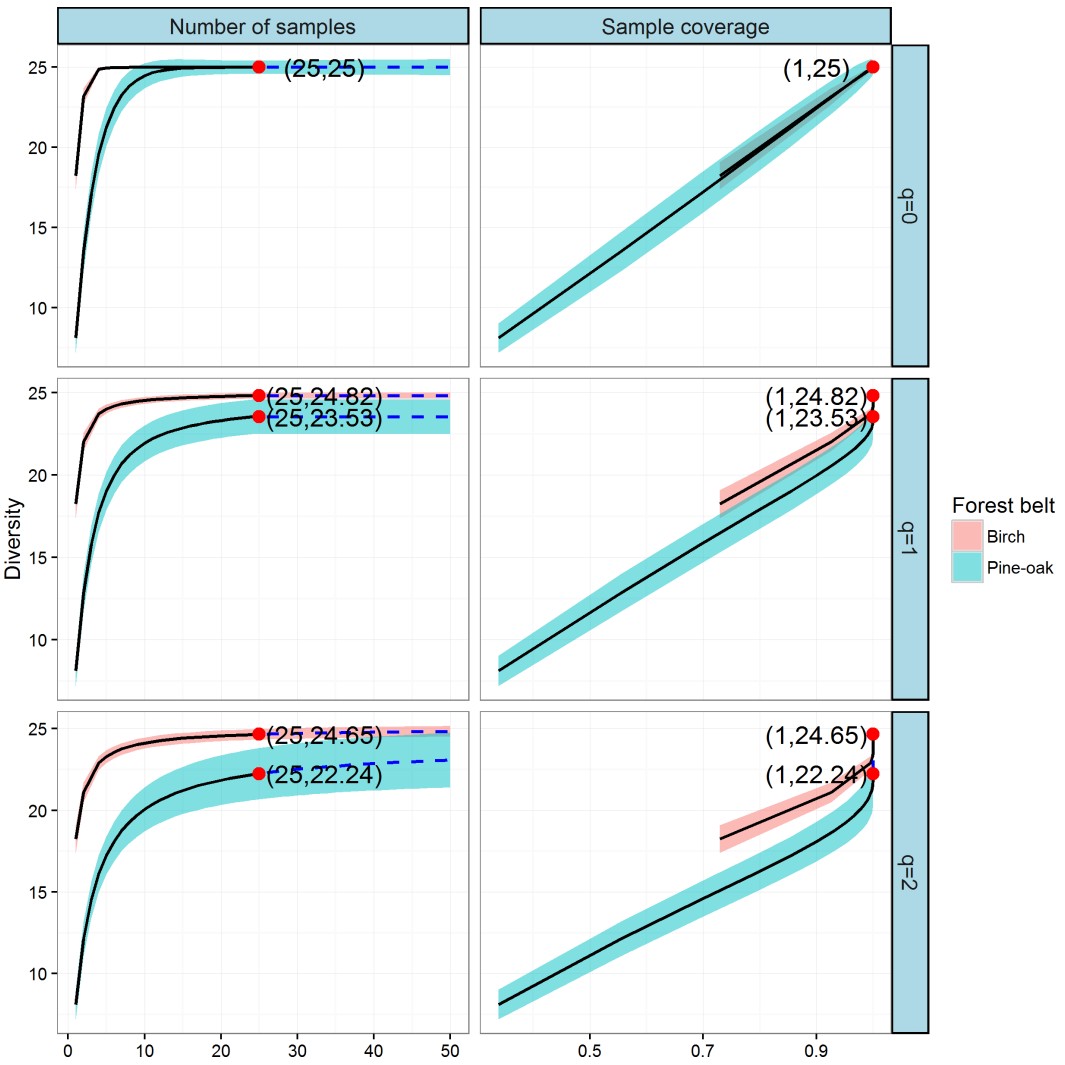

**Figure 6  Sample-based and coverage-based rarefaction and extrapolation curves based on the Hill numbers ($q = 0, 1, 2$) for the birch and pine-oak belts.** The 95% confidence intervals (shaded regions) were obtained by a bootstrap method based on 200 replications. Reference samples are denoted by solid dots, the numbers in the parentheses are the sample size and the observed Hill number for each reference sample.

Simpson diversities for the birch and pine-oak belts, implying that the birch belt was more diverse than the pine-oak belt. The confidence intervals for species richness, however, overlapped considerably between the birch and pine-oak belts.

## Distribution of species abundance

The observed SADs of the tree communities of the birch and pine-oak belts, together with the distributions fitted by the six classical models (broken-stick, niche-preemption, log-normal, Zipf, Zipf-Mandelbrot, and neutral-theory), are shown in Fig. 7. The expected and observed SADs of the birch belt differed significantly (indicated by a K-S test). The niche-preemption, neutral-theory, broken-stick, and log-normal models simulated

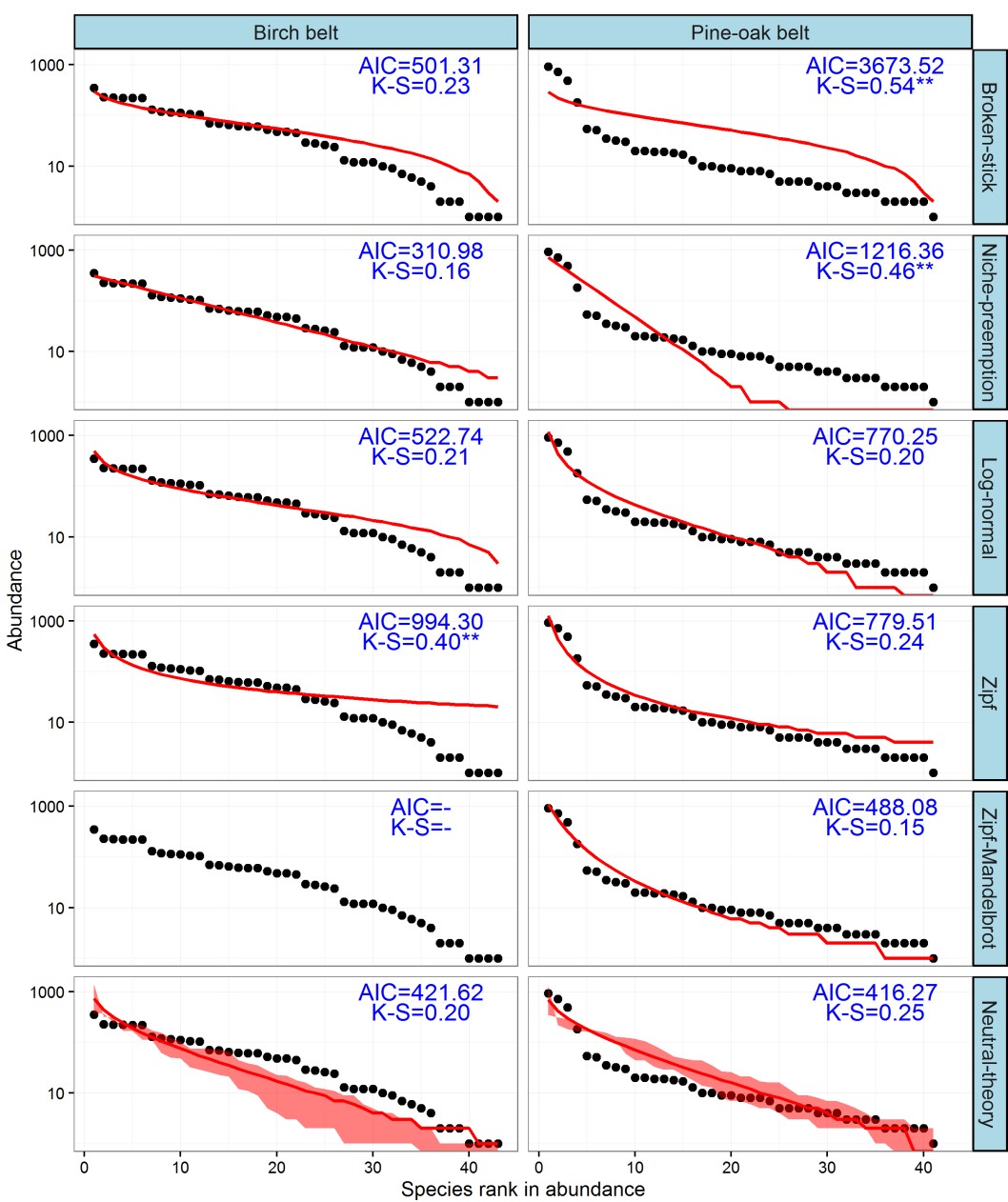

**Figure 7** **Species-abundance distribution and model fittings of the typical secondary forests for the birch and pine-oak belts in the mid-altitude zoneof the Qinling Mountains, China.** The Zipf-Mandelbrot model failed to fit the data of birch belt. AIC, Akaike's Information Criterion; K-S, statistic of the Kolmogorov-Smirnov test. ***, $P < 0.001$; **, $P < 0.01$; *, $P < 0.05$.

SAD of the birch belt well, but the observed SAD departed from the outputs of the Zipf-Mandelbrot and Zipf models; the Zipf-Mandelbrot model especially failed to fit the SAD patterns. Among the models, the niche-preemption had lowest AIC value, indicating that this model best represented the SAD pattern of the birch belt. The neutral-theory, Zipf-Mandelbrot, log-normal, and Zipf models simulated SAD well for the pine-oak belt, and the neutral-theory model were best for the SAD patterns.

## DISCUSSION

### Tree species composition and community structure

The birch and pine-oak belts had rich species compositions and similar floristic components, but the tree community structures clearly differed. The mid-altitude zone in the Qinling Mountains is rich in forest resources and species diversity that provide an important gene pool (*Lei, Peng & Chen, 1996*; *Wang et al., 2015*). Birch and pine-oak belts are the two main forest types in the zone (*Liu et al., 2001*; *Zhao, Ma & Xiao, 2014*), with rich species compositions (*Lei et al., 1996a*; *Lei et al., 1996b*; *Wang et al., 2015*; *Zhang et al., 2014*), in agreement with our findings. The numbers of species, genera, and families are very similar between the belts (Tables 4, 5 and the species composition and IV characteristics in Supplemental Information 4), perhaps due to the similarity of their habitats. The range of the mid-altitude zone (1,300–2,600 m) is relatively small, especially in our study forests distributed between 1,400 and 2,400 m, so altitude would have little effect on species distribution and composition, and these two forest belts share most species of trees and have similar floristic components. Both the cluster and correspondence analyses, however, demonstrated a clear difference between the belts. Previous studies have shown that climate change (*Zhao, Ma & Xiao, 2014*), the influence of species interaction on the pattern of floristic composition, small-scale topographic variation, and soil conditions (*Lei, Peng & Chen, 1996*; *Lei et al., 1996a*; *Ren et al., 2012*; *Wu et al., 2012*; *Zhao et al., 2003*) among forest stands can affect the distribution of forest stands in the mid-altitude zone of the Qinling Mountains.

### Tree diversity patterns

Tree diversity was significantly higher in the birch than the pine-oak belt. The confidence intervals for species richness ($q = 0$) overlapped considerably between the birch and pine-oak belts, but the rarefaction curves for Hill numbers ($q = 1, 2$) indicated that tree diversity was significantly higher in the birch than the pine-oak belt; inferences for diversities of $q \geq 1$ are reliable (*Chao et al., 2014*). Tree diversity was significantly higher in the birch than the pine-oak belt, likely for two main reasons. (1) The distributional range suited the birch belt better. *Lei, Peng & Chen (1996)* reported that the birch belt was distributed toward the upper limit of the mid-altitude zone (1,800-2,400 m) in the study area, and the pine-oak belt was distributed at lower elevations (1,200-2,000 m); species richness and diversity, however, were highest between 1,800 and 2,200 m, because the elevation zone is an ecotone of birch and pine-oak belts. Diversity may also have been higher in the birch than the pine-oak belt because the elevation zone (1,800–2,200 m) was within the main distributed zone of birch belts. Other studies have supported this proposal that species diversity in the Qinling Mountains is higher between 1,800 and 2,200 than in other elevation zones (*Kang & Zhu, 2007*; *Tang, Fang & Zhang, 2004*; *Xin et al., 2011*). (2) These two belts were the most common forest types, but the dominance of dominant species differed between the belts as the forests developed. The dominant species *B. albo sinensis* was not very conspicuous in the birch belt, and pure stands were rare (*Lei et al., 1996a*). In contrast, *P. tabuliformis*, *P. armandii*, and *Q. aliena* var. *acutiserrata* predominated in the pine-oak belt (*Liu et al., 2001*). These dominances were reflected

by the *IV* index (See the species composition and IV characteristics in Supplemental Information 4). *IV* was highest for *B. albo sinensis* in the birch belt (10.63%) but only slightly higher than for the other dominant tree species. The *IVs* of the predominant species *Q. aliena* var. *acutiserrata* (*IV* = 26.15%), *P. tabuliformis* (*IV* = 20.05%), and *P. armandii* (*IV* = 22.50%) in the pine-oak belt indicated evident advantages.

## Mechanism of coexistence of tree communities

Niche and neutral processes are simultaneously influencing the distribution of species and the community dynamics of the birch and pine-oak belts. The neutral-theory model was suited to the data for species abundance for both belts, which identified randomness as the main ecological process determining the distributional pattern of species abundance in these two forest belts. These forests can thus maintain a dynamic balance during growth and development and are amenable to stable development, supporting the findings by *Lei, Peng & Chen (1996)*, *Lei et al. (1996a)* and *Lei et al. (1996b)*. The log-normal model had good predictive power for the SAD patterns of both belts, which further confirmed that statistical models based on statistical theory (e.g., the log-normal model) are superior to resource-apportioning models based on ecological theory (e.g., the broken-stick and niche-preemption models) (*McGill et al., 2007*).

The niche-preemption model and broken-stick model were also suitable for simulating SAD patterns for the birch belt, which showed that niche theory was important in the community assemblages of the birch belt. *Lei et al. (1996a)* reported that the dominant species *B. albo sinensis* regenerated poorly in the Qinling Mountains and that the continuity of *B. albo sinens* populations was maintained by gap regeneration. These findings are consistent with the regeneration-niche hypothesis (*Grubb, 1977*) and suggest that both the neutral and niche theories have played important roles in understanding the mechanisms of species coexistence in the birch belt.

The combination of the Zipf/Zipf-Mandelbrot (niche-based model) and neutral-theory models suggested that the pine-oak belt contains progressive successional communities and can maintain stable community development during succession, consistent with the findings by *Chai & Wang (2015)* and *Lei, Peng & Chen (1996)*. We concluded that the successional characteristics of pine-oak forests accords with the ecological interpretations of the Zipf/Zipf-Mandelbrot model that climax species need more time and resources to replace the pioneer species during succession but ultimately survive for a long time. Species of pines are common pioneer species and are often later succeeded by climax species of oaks (*Gracia, Retana & Roig, 2002*; *Yu et al., 2013*; *Broncano, Riba & Retana, 1998*), and pine-oak mixed forests are usually an initial successional stage after a disturbance in pine forests where pines mainly dominate the forest canopy and oaks predominate in the understory (*Gracia, Retana & Roig, 2002*; *Yu et al., 2013*). Our results support this successional series, and our previous observations and studies also suggest that pine-oak mixed forests become oak forests within a few decades in the Qinling Mountains (*Kang, Wang & Cui, 2011*; *Xu, 1990*; *Yu et al., 2013*).

Many studies have warned against drawing conclusions based on the ability of exclusive models to fit SAD patterns (*Chen, 2014*), because the data may be equally well fitted by more

than one model, which may provide substantially different interpretations. Our results at least suggest a possibility that niche and neutral processes are simultaneously influencing the distribution of species and the community dynamics of the birch and pine-oak belts. Both the findings by *Legendre et al. (2009)* for a subtropical evergreen broadleaved forest at the Gutianshan National Nature Reserve in eastern China and by *Zhang, Zhao & Von Gadow (2010)* for a temperate forest at Changbaishan in northeastern China also indicated that niche and neutral processes were simultaneously regulating species coexistence.

## Conclusion and recommendations

The conservation and management should receive more attention to protect biodiversity and the forest resources in the Qinling Mountains. Understanding forest species composition, diversity patterns, and community assemblages are very important for managing ecosystems for their environmental and conservation value (*Jung et al., 2014*; *Kacholi, Whitbread & Worbes, 2015*; *Ragavan et al., 2015*). Protecting biodiversity and forest resources in the Qinling Mountains has become a focus of attention (*Lei, Peng & Chen, 1996*; *Wang et al., 2015*; *Zhao, Ma & Xiao, 2014*). Although the forests with rich species compositions and many forest fragments remain at risk (*Cheng et al., 2015*; *Wang et al., 2015*), their conservation must be given priority to avoid the loss of species, especially endemic and nearly endemic species. In addition, the forests have been harvested since the 1950s, and much of the area is now covered by secondary growth that has low productivity and poor community stability and with varying patterns of natural succession (*Chai & Wang, 2015*; *Cheng et al., 2015*; *Li, Ji & Liu, 2004*). Enhancing the multi-functionality of forests is a goal of modern and sustainable forest management, which tries to balance a multitude of economic, ecological, and societal demands. Increasing the tree diversity of forests is particularly promising (*Schuldt & Scherer-Lorenzen, 2014*). We suggest that scientific management of the forests should be increased to improve forest quality and productivity and consequently to realize the sustainable use of the forest resources.

## ACKNOWLEDGEMENTS

The authors thank Huinan Zhu and Dr. Fei Yu for providing the photos used in this article, and also thank the Qinling National Forest Ecosystem Research Station at Huoditang, Ningshan County, Shannxi Province, for its strong support of the field investigation.

### Funding

This study was funded by National Natural Science Funds of China (Grant No. 31470644), and CFERN & GENE Award Funds on Ecological Paper. The funders had no role in study design, data collection and analysis, decision to publish, or preparation of the manuscript.

### Grant Disclosures

The following grant information was disclosed by the authors:
National Natural Science Funds of China: 31470644.
CFERN & GENE Award Funds on Ecological Paper.

## Competing Interests

The authors declare there are no competing interests.

## Author Contributions

- Zongzheng Chai conceived and designed the experiments, performed the experiments, analyzed the data, contributed reagents/materials/analysis tools, wrote the paper, prepared figures and/or tables, reviewed drafts of the paper.
- Dexiang Wang conceived and designed the experiments, contributed reagents/materials/analysis tools, wrote the paper, reviewed drafts of the paper.

## Data Availability

Raw data can be found in the Supplemental Information.

## Supplemental Information

Supplemental information for this article can be found online at http://dx.doi.org/10.7717/peerj.1900#supplemental-information.

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
