# Peer review of "A comparison of species composition and community assemblage of secondary forests between the birch and pine-oak belts in the mid-altitude zone of the Qinling Mountains, China"

_PeerJ, doi:10.7717/peerj.1900_

## Round 0.1 · original submission · Major Revisions

This is a soild manuscript describing composition and diversity of species assemblages of two forest types in the Qinling Mountains of China. The methods are generally sound and the work is placed in an appropriate broader context.

In addition to addressing in your revision the comments of the three reviewers, I have a number of additional comments and suggestions.

1. Although "novelty" is not a requirement for papers published in PeerJ, there is also no need for extraneous discussion of unrelated issues. Primarily, it is not clear why "monitoring biodiversity is essential for the recovery and development" of these forest resources. The forest will recover and develop as long as it is left alone, and monitoring its biodiversity will not help it along. As suggested by Reviewer 3, the results from the study do not apply to forest restoration. Rather than making this linkage, I agree with Reviewer 3 that you can drop this claim.

2. In the Methods, you describe in detail the Mingling index for species diversity, but not the others in detail. Since all are described well in the Supplemental file, I think you can eliminate Lines 151-176 in the main text as long as that material is in the Supplemental file.

3. The confidence intervals around the species accumulation curves (Fig. 4) should not close off at the maximum number of plots. This is an artefact of earlier software, and has been resolved in new releases of software such as EstimateS and SPADE. Please recalculate these and redraw them appropriately.in Fig. 4.

4. As noted by Reviewer 1, there are some discrepanceis between your text, Figure 5, and Table 7. Please recheck these for consistency and revise the ms. appropriately.

5. What do the letters mean in Figure 6, and how were these presumed differences calculated? ANOVA? Please specify in the Methods.

6. Divseristy indices 1-4 should be recalculated as "Hill numbers" (see Chao et al. 2014, Ecological Monographs 84: 45-67 for details).

7. As noted by Reviewer 3, Figure 7 should be presented first in the Results, and the methods used to create them placed in the Methods.

8. Reviewer 2 makes important points about different types of diversity, especially beta diversity and sampling effort. These are important to address.

9. The inset in Figure 3A is not described and is not obviously needed. Please delete it. The other colors in Fig. 3A besides the magenta for Qinling and the red point for Huoditang are not needed. Please delete them as well.

10. Please recheck the next version for English language and style, and punctuation. The ms. is generally clearly written, but could use one more round of careful proofreading.

I look forward to seeing a revision in due course.

Reviewer 1 ·

Basic reporting

This manuscript deals with the community assemblage and diversity of two different forests belts which have been recovered from deforestation. The investigated belts located at the mid-altitude zone of the Qinling Mountains in China. This region is very important as the water supply for a huge number of people and as a refuge for many species. Any disturbance of this system may have an ecological impact. Therefore studies on the community assemblage and diversity of secondary forests are of high interest.
However I have some major problems with the present data, data analysis and presentation.
1. The English of your manuscript must be improved.
2. Please better identify the novelty of your work.
3. This study randomly selected 25 plots for each belt which have the same stand age, however, the elevation, slope and aspects, which have not shown in the manuscript, would create topoclimates in mountain regions and have significant effect on the community diversity. Here, the range of the mid-altitude zone (1400-2400 m) is relatively high, thus, the comparison between the two forest belts under such circumstance seems not reasonable.
4. There are some errors when interpreting the picture 5 in the result section.
In lines 211-216 “The niche-preemption, neutral-theory, broken-stick, and log-normal models simulated SAD of the birch belt well. The observed SAD departed from the outputs of the Zipf-Mandelbrot and Zipf models (Figure 5, Table 7). The niche-preemption and neutral-theory models were much superior to the other models and should be suitable for simulating SAD patterns for birch belts.”
However, in Figure 5a, we found that niche-preemption and Zipf-Mandelbrot models are better, and neutral-theory model do not fit well. Furthermore, the niche-preemption model and Zipf-Mandelbrot model have the same fitting effect; however, the values of K-S are significant difference, so please explain the reason?
Similar mistake also found in the Figure 5b, line 215 “Zipf-Mandelbrot and neutral-theory models were better suited to the SAD patterns of the pine-oak belt”. However, we found that the log-normal and Zipf-Mandelbrot models were better.
5. The last paragraph in discussion section is not closely related to topic in this MS. Please, revise of remove.

Experimental design

There are no scientifc questions or hypothesis. Why did the authors perform this research?

Validity of the findings

It is better to identify the novelty of your work. If the author can not show the novelty, it can not be published.

Additional comments

It is not well written, and the English writing needs to be checked by a native English speaker.

Reviewer 2 ·

Basic reporting

This paper is very descriptive, however I fail to see a hypothesis. Yes - the two forest types are different - but why is this important? Is the purpose of this paper just to inventory biodiversity? If so, then I think that there should be more interesting hypotheses included in the paper.
I also understand that a major part of this paper was to look into community establishment, however I also fail to see the importance of this based exclusively on what was outlined in this paper alone. I feel like this paper is very reliant on past experiments (i.e. references), but to an extreme, to where I cannot unravel how this paper helps improve our understanding of the biodiversity of the Qinling Mountains.

Experimental design

I think that the size of the plots are quite impressive. However, my biggest concern is with the Species Accumulation Curves developed for each forest belt. It is clear in your manuscript that the SAC for the Pine-Oak belt was undersaturated. This means that your sample size was undersaturated as well. Therefore, I think that it is difficult to draw any conclusions about comparative diversity metrics if you data cannot account for a more accurate representation of the tree diversity levels in the Pine-Oak forest belt.

Validity of the findings

The statistical methodology used in this paper seems sound, however I am not that well-versed in many of the metrics used (i.e. broken-stick, Mingling, and Zipf-Mandel.) Therefore, I cannot give sound judgement on the validity of many of the findings because on my lack of knowledge of the metrics.

There were a few areas that I found quite suspect, mainly when comparing the diversity levels of the two forest belts. In the paper, the authors conclude that the birch belt has a statistically higher diversity than the pine-oak belt, however the reported gamma levels of each belt were 43 and 41 species, respectively. Therefore, I would be more interested in knowing about the heterogeneity of the system (i.e. beta diversity) than the reports of the relative alpha diversity levels in the area since the gamma diversities are pretty much the same. I also think that, since the oak-pine belt was undersampled, an increase in sample size might shed light onto the variability of tree diversity both within the pine-oak belt and in its relation to the birch belt forest as well.

Additional comments

In terms of grammar, spelling, and overall syntax, I felt that the paper was well-written. I just wish that there had been a more compelling and interesting story to go along with the written aspects of the paper.

·

Basic reporting

Overall, this is a clearly written paper. It is a solid contribution to the literature on forest community ecology. In particular, it is a nice extension of the neutral and niche theories of species coexistence to a species-rich forest outside of the tropics. The figures are easy to interpret and good quality. The raw data file appears to be complete.

There are some modifications in the technical terminology needed prior to publication
• Please use ‘ha’ rather than ‘hm2’ as the abbreviation for hectare
• The term ‘deforestation’ to me implies a conversion to non-forest. Timber harvesting, even as a clearcut, is not a conversion to another land-use, and thus I’d prefer that the forest studied here be termed ‘second growth,’ ‘post-harvest,’ or ‘secondary succession.’
• Please define the term “constructive species” (first used line 243); this isn’t a common enough term to assume the readership will understand its meaning.
• Similarly, consociation (line 274) has a specialized meaning that many readers may not know. Either re-phrase (maybe monospecific or single-species) or define.

Introduction
• Line 45. By ‘original forest’ do you mean never-harvested?
• Line 45. ‘After years of protection’ – does this mean that forest regeneration was delayed or required special intervention? That is, after timber harvesting, are restoration measures necessary for a secondary forest to regenerate? Without restoration, does the forest convert to non-forest? I think that making the timeline and any human intervention required to secure secondary growth in this forest is key to readers understanding how the results of the study apply to restoration work.

Methods
• Be sure to include the cluster analysis and CA analysis methods here. Define what CA analysis is. Don’t just introduce these in the Discussion.

Results
• Figure 7 should be presented first in the Results. It can be referred to again in the Discussion, but shouldn’t be presented there. In fact, Figure 7 is the most direct test of the stated goal of comparing the birch vs. oak-pine forests. The other figures and tables are interesting descriptors of each community’s composition and internal diversity, but the cluster analysis and ordination directly show the dissimilarity between the two major community types.

Discussion
• Line 237. Is randomness an ecological process, or a lack of one?

Experimental design

• The methodology and analysis is sound. However, I think the study question needs to be refined to match the conclusions that the authors can draw from the study. In particular, it isn’t clear how the results from this study apply directly to forest restoration. Either make this link more explicit, or drop this claim. The paper is fine without it.

• It is unclear to me whether the set of plots analyzed in this paper are new, or if they have been used in prior publications to explore other questions. If the former, please state this. If the latter, please clearly note the prior publications resulting from their measurements.

• Similarly, please add which year(s) the plots were sampled.

Validity of the findings

The analyses and conclusions appear to be statistically sound, and appropriate for the data set. There is one gap in the context that needs to be addressed:
• The Introduction fails to cite prior work comparing secondary to primary forest in the Qinling region. It is therefore unclear whether the forests in this study have low diversity relative to the original forest, or if their species richness and assemblages are similar to primary forest.

Additional comments

No additional comments.

---

## Round 0.2 · accepted · Accept

Thank you for doing a thorough revision of this manuscript. It is a really nice contribution.

***Please check the legend for Figure 4. The figure is a cluster diagram and ordination, not a species-accumulation curve.***